# Effect of Prism Adaptation Therapy on the Activities of Daily Living and Awareness for Spatial Neglect: A Secondary Analysis of the Randomized, Controlled Trial

**DOI:** 10.3390/brainsci11030347

**Published:** 2021-03-09

**Authors:** Katsuhiro Mizuno, Kengo Tsujimoto, Tetsuya Tsuji

**Affiliations:** 1Department of Physical Rehabilitation, National Center Hospital, National Center of Neurology and Psychiatry, 4-1-1 Ogawa-Higashi, Kodaira, Tokyo 187-8551, Japan; tsujimoto_k@ncnp.go.jp; 2Department of Rehabilitation Medicine, Keio University School of Medicine, 35 Shinanomachi, Shinjuku-ku, Tokyo 160-8582, Japan; cxa01423@nifty.com

**Keywords:** rehabilitation outcome, hemispatial neglect, behavioral assessment, visuomotor adaptation, anosognosia for spatial neglect, cognitive rehabilitation

## Abstract

Background: Rehabilitation for unilateral spatial neglect (USN) using prism adaptation (PA) is one of the most widely used methods, and the effectiveness of PA is well-evidenced. Although the effect of PA generalized various neglect symptoms, the effectiveness for some aspects of neglect is not fully proven. The Catherine Bergego Scale (CBS) was developed to identify problems with the activities of daily living (ADL) caused by USN. The CBS is composed of 10 observation assessments and a self-assessment questionnaire. To assess the self-awareness of USN, the anosognosia score is calculated as the difference between the observational scores and the self-assessment scores. To investigate how PA affects ADL and self-awareness in subacute USN patients during rehabilitation, we analyzed each item of the CBS and self-awareness from a randomized, controlled trial (RCT) that we previously conducted (Mizuno et al., 2011). Methods: A double-masked randomized, controlled trial was conducted to evaluate the effects of a 2-week PA therapy on USN in 8 hospitals in Japan. We compared each item of the CBS, anosognosia score, and absolute value of the anosognosia score between the prism group and the control group. Results: Two of ten items (gaze orientation and exploration of personal belongings) were significantly improved in the prism group compared with those in the control group. The absolute value of the anosognosia score was significantly improved by PA. Conclusions: Improvement of oculomotor exploration by PA may generalize the behavioral level in a daily living environment. This study suggested that PA could accelerate the self-awareness of neglect during subacute rehabilitation.

## 1. Introduction

Unilateral spatial neglect (USN) can be defined as a failure to orient to contralesional stimuli, when this failure cannot be attributed to either sensory or motor defects [1]. It has been reported that USN occurs in approximately 50% of right brain-damaged and 30% of left brain-damaged stroke patients [2,3]. Because the most important clinical problem in USN patients is interference with rehabilitation processes resulting in poor functional outcomes [3,4,5,6,7,8,9,10,11,12,13], various rehabilitation techniques for USN have been explored [14,15].

Prism adaptation (PA) therapy was first described by Rossetti et al. [16]. During prism adaptation, individuals repeatedly perform goal-directed pointing movements wearing glasses or goggles with prism lenses that induce a rightward or leftward optical shift [17]. While the initial movements are directed to the visually shifted position of the target, the pointing error rapidly decreases after subsequent pointing movements. After the prisms are removed, a pointing error opposite to the prismatic shift, so-called “after-effect,” is observed. Rossetti et al. [16] demonstrated a significant reduction in the spatial neglect following a brief period of PA with rightward prisms. In later studies, it was confirmed that the effects of PA could be generalized across the various clinical aspects of USN and other cognitive disorders [18,19]. Previously, we conducted a randomized controlled trial in 2011. It was reported that a 2-week PA therapy and following conventional rehabilitation significantly improved the rehabilitation outcome of ADLs in subacute stroke patients with USN [20]. However, this study failed to demonstrate a significant time-group interaction between the intervention group and the control group in either the behavioral inattention test (BIT) [21] or the Catherine Bergego Scale [22], which are direct measures of USN.

It has been suggested that the effect of PA generalized the various levels of neglect symptoms, such as oculomotor exploration, visuo-verbal tasks, and ADL [18]. Furthermore, a recent systematic review [18] suggested that PA could improve reading/writing and the ADL direct test (for example, BIT behavioral test score and CBS score). However, several previous studies suggested that several aspects of USN could not be improved by PA [23,24]. Ferber et al. [23] reported a case with USN in which PA improved the exploratory eye movement but not facial discrimination in the left hemispace. In addition, Gossmann et al. [24] reported that PA improved viewer-centered neglect but not stimulus-centered neglect. However, it is not well understood whether these homogenous effects of PA are reflected in ADL.

The CBS [22] was developed to identify problems with ADL caused by USN symptoms. The CBS is composed of 10 observation assessments of ADL, such as grooming, dressing, and eating. The psychometric properties of these assessments have been reported, and they have a good reliability, validity, sensitivity, and responsiveness during rehabilitation [22]. Therefore, the CBS may be the most widely used behavioral assessment instrument for USN [25]. To evaluate self-awareness for neglect, a self-evaluation questionnaire on the same 10 items of the CBS were designed. An anosognosia score was defined as the difference between the observational scores and the self-assessment scores. If the observational score is greater than the self-assessment score, the anosognosia score is positive. Positive values of the anosognosia score indicate the unawareness of USN. The unawareness of USN was related to the severity of the unilateral neglect [22,26]. Vocat et al. [27] reported that the observational scores and self-assessment scores of the CBS in patients with right hemisphere lesions are not correlated in the acute phase, while they are strongly correlated in the chronic phase. These results suggested that most patients with USN were unaware of their neglect in the acute phase, and both their neglect and awareness were improved in the chronic phase. However, some patients with moderate to severe USN indicated a negative value of the anosognosia score [22]. During the recovery process, the overestimation of neglect is likely to occur to compensate the various difficulties of ADL caused by neglect. Therefore, a negative value as well as a positive value of the anosognosia score may indicate a discrepancy between actual disability and self-estimated disability. To test the hypothesis, the absolute value of the anosognosia scores should be analyzed in patients undergoing rehabilitation for USN in the subacute to chronic phase. In Japan, the time between stroke onset and admission to a rehabilitation hospital is 3–5 weeks on average, and the average length of hospital stay is 3–4 months. Therefore, we consider this time as the sub-acute phase. Furthermore, the effects of PA on the awareness of neglect during the subacute rehabilitation period should be investigated.

In our previous RCT, we measured the observational and self-assessment scores of the CBS [20]. However, we did not analyze each item and the anosognosia score of the CBS in detail. In this study, which is a secondary analysis of our past RCT [20], (1) each item of the CBS was compared between the prism group and the control group to investigate how PA improves neglect symptoms in ADL, and (2) changes in the anosognosia score and its absolute value on the CBS were examined throughout the subacute rehabilitation period in both groups to clarify the effect of PA on the awareness of neglect.

## 2. Materials and Methods

### 2.1. Participants, Randomization, Intervention, and Data Collection

Details of the study protocol were published in the previous study [20]. We enrolled right hemisphere damaged stroke patients with USN within 3 months after onset. USN was diagnosed by at least 1 item scoring less than the cutoff value in the standard test of the BIT. All patients provided written informed consent. Participants were randomly allocated to the intervention or the control group, with a computerized block randomization scheme. Patients in the prism group performed a repetitive pointing task under a table to hide their hand trajectories 90 times with prism glasses that shifted their visual field 12° to the right (Fresnel lens: Koyo Corporation, Tokyo, Japan). Patients underwent 2 daily sessions, 5 days a week for 2 weeks, for a total of 20 sessions. Those in the control group performed the same task with neutral glasses. The evaluation was conducted at the start of the study (T0 = baseline), immediately after 2 weeks of intervention (T1 = after treatment), and at discharge (T2 = follow-up). 

### 2.2. Catherine Bergego Scale

The CBS is a functional neglect assessment that also has a parallel questionnaire, administered to stroke survivors, to determine whether they can self-assess neglect [22]. The CBS is based on the direct observation of patient function in 10 actual situations. For each item, a 4-point scale ranging from 0 (no ignore) to 3 (severe ignore) was used. If no spatial bias was observed, a score of 0 was given. For mild neglect, a score of 1 was given, and the patient always searched the right hemispace first, slowly to the left, and occasionally showed a left-sided loss. A score of 2 (moderate neglect) was given if the patient showed a clear and constant left dropout or collision. A score of 3 (severe neglect) was given when the patient was not able to fully explore the left hemispace. The total score was calculated (range, 0–30). For items that could not be measured, the average score of the measured items was entered. In this study, the CBS scores were assigned by an occupational therapist.

An anosognosia score was calculated by recording the difference between the observer’s assessment scores and the patient’s self-assessment scores. The anosognosia score is of a positive value if the patient has a higher CBS score than the rater. This means that patients are unaware of their problems caused by USN. However, a negative value means that patients overestimate their spatial problems. Both of these can be considered invalid self-assessments. In this study, both the usual anosognosia score and the absolute value of the anosognosia score were calculated.

### 2.3. Statistical Analysis

Data was analyzed with MATLAB (R2019a). Each item of the CBS at each time point (T0, T1, and T2) was compared between the prism and control groups by the Mann–Whitney U test. Correlations between the CBS score and anosognosia score (and absolute value of anosognosia score) at each time point were evaluated with the Spearman’s rank correlation test. Analysis of variance (ANOVA) was conducted to analyze the differences in the CBS, anosognosia score, and absolute value of the anosognosia score between the two factors of time (T0, T1, and T2) and group (control and prism). Post-hoc multiple comparison with Bonferroni’s correction with Bonferroni multiplicity adjustment was performed. For all tests, *p* values less than 0.05 were considered significant. 

## 3. Results

### 3.1. Group Characteristics

Baseline characteristics of the two groups are shown in Table 1. No significant differences between the prism group and control group were found.

### 3.2. Comparison of Each Item of the CBS between the Prism Group and Control Group

Table 2 indicates the average scores of each item of the CBS at each time point. Scores of gaze orientation and the personal belongings were significantly lower in the prism group than in the control group at T2. The other items were not significantly different between the prism and control groups at each time point.

### 3.3. Correlation among the Observational Score of the CBS, Anosognosia Score, and Absolute Value of the Anosognosia Score

There was a strong correlation between the CBS and the anosognosia score at T0 and a moderate correlation at T1, but no correlation was found at T2 (Figure 1A). However, there was a strong correlation between the CBS and absolute value of the anosognosia score at all time points (Figure 1B). 

### 3.4. Effect of PA on the CBS Score, Anosognosia Score, and Absolute Value of the Anosognosia Score

Table 3 shows average of CBS sore, anosognosia score, and absolute value of anosognosia score at each time point. 

With a two-factor ANOVA, there were significant effects of time in the CBS score (F(5,72) = 7.557, *p* < 0.001). However, the 2-way Group × Time interaction was not significant. The CBS score was significantly lower at T1 and T2 than at T0 in both groups (Figure 2A).

In the anosognosia score, there were significant effects of time with a two-factor ANOVA (F(5,72) = 14.143, *p* < 0.001). However, the 2-way Group × Time interaction was not significant. The anosognosia score was significantly lower at T1 and T2 than at T0 in both groups, and it was significantly lower in the control group at T2 (Figure 2B). However, the average score of the prism group was close to zero and that of the control group was slightly negative.

In the absolute value of the anosognosia score, there were significant effects of time with a two-factor ANOVA (F(5,72) = 5.688, *p* < 0.001). However, the two-way Group × Time interaction was not significant. The absolute value of the anosognosia score was significantly lower at T1 and T2 than at T0 only in the prism group, and it was significantly lower in the prism group at T2 (Figure 2B).

## 4. Discussion

This study is the secondary analysis of a previous RCT to investigate the effectiveness of PA therapy on subacute stroke patients with USN. The study suggested that PA facilitated motor and cognitive re-learning after the two-week treatment and improved the rehabilitation outcome of ADL in the subacute rehabilitation period. In addition, this secondary analysis suggested that PA could improve visual exploration in daily life and facilitates self-evaluation for difficulties in ADL.

In this study, gaze orientation and personal belongings, among the 10 items of the CBS, were improved in the prism group compared with the control group. Previous studies suggested that PA can ameliorate eye movement [23,28] and viewer-centered neglect [24]. These abilities are helpful to oculomotor exploration in daily living related to such two items of the CBS. In addition, it has been reported that PA strongly affects premotor neglect [29,30,31,32]. Patients with premotor neglect have difficulty exploring personal belongings in the contralesional hemispace because of reaching bias to the ipsilesional hemispace. If PA could improve premotor neglect, the patients could explore personal belongings in the contralesional hemispace more easily. Furthermore, it is commonly observed that patients with neglect often re-inspect the space that they have already inspected [33]. For example, when left-sided USN patients perform a cancellation task, some of them repeatedly check rightward targets that they have already found. It is considered that this perseveration error could be caused by an impaired visual working memory. In the settings of ADL, patients with visual working memory deficits may tend to explore the same place many times when they explore personal belongings. Previous studies suggested that PA may lead to the amelioration of the disrupted spatial working memory in patients with USN [34,35]. If spatial working memory deficits are ameliorated, they no longer explore the space that has been explored. Therefore, we hypothesized that the exploration of personal belongings was improved, and this was caused by (1) an improvement in the premotor component of neglect, (2) an improvement in gaze shifts, (3) an improvement in viewer-centered neglect, and (4) an improvement in visual working memory.

During the rehabilitation procedure, patients progressed from initial unawareness to emergent/online awareness in their difficulties caused by USN [36]. This emergent/online awareness is proposed by the patient’s experience of difficulties in various daily activities. However, offline awareness is based on the memory of patients. The anosognosia score of the CBS is considered as offline awareness. Chen and Toglia [37] reported that the severity of offline awareness was not correlated with neglect severity, while online awareness was significantly correlated with neglect severity in subacute stroke patients about 1 month after the onset of stroke on an average. Furthermore, it was indicated that 21.9% of no neglect patients complained regarding difficulties related to a spatial problem. These results indicated that the overestimation of neglect is likely to occur in the assessment of offline awareness like in the self-evaluation of the CBS. In this study, both the anosognosia score and its absolute value were strongly correlated with the CBS observational score at baseline. However, at discharge, only the absolute value was correlated with the CBS. In addition, the absolute value of the anosognosia score had a stronger correlation with the CBS than the anosognosia score throughout the study period. Therefore, the absolute value is considered to be a better reflection of emergent/online awareness than the relative value. In the case of patients undergoing rehabilitation, it may be important to consider the distance between self-assessment and observation, rather than simply the difference between them.

There was no significant difference between the prism and control groups in the CBS score from T0 to T2. The lack of difference between the prism and control groups can be attributed to three factors, as discussed in the original paper: (1) the differences are masked because spontaneous recovery might have occurred in both groups; (2) repetition of the tests and conventional rehabilitation might have caused a treatment effect in the control group; and (3) there might have been ceiling effects of the outcome measures because most of the patients had mild to moderate CBS scores. However, because significant improvement was observed in ADL [20], prism adaptation might boost the rehabilitation process.

There were significant differences in the absolute values of the anosognosia score at T1 and T2 compared with that at T0 only in the prism group. Moreover, it was significantly lower in the prism group than in the control group at T2. These results suggested that PA could improve emergent awareness for spatial problems. Previous studies reported that unawareness of neglect was the most important predictor of performance in standardized ADL [38], and the severity of neglect at three months and six months post-stroke could be predicted by the presence of unawareness in the acute phase [39]. In other words, the early recognition of the patient’s own condition is probably the most important factor in the subsequent improvement of ADL. In the PA procedure, patients experience visuo-motor mismatch induced by prism glasses. This experience may facilitate to obtain the emergent awareness of neglect. Therefore, it may be hypothesized that PA therapy improved emergent awareness for neglect and facilitated the improvement of ADL by following the rehabilitation procedure.

A limitation of this study is that we did not investigate whether the lesion location and neural mechanisms affect self-assessment and the self-assessment recovery of neglect. Studies indicate that there are regions of the brain that are associated with the different types of self-assessment [40,41,42]. Neuroanatomical relationships to neglect subtypes (e.g., personal, peripersonal) have also been identified [43]. Especially, the regions and fibers of the middle frontal gyrus, inferior frontal gyrus, and superior longitudinal fascicle III were reported to be associated with self-assessment [44]. In addition, we have not been able to analyze the correlation between the anosognosia score and the improvement of neglect by PA. Awareness of neglect may be a candidate to predict a better outcome by PA therapy. Therefore, it is necessary to examine these limitations in the future.

## 5. Conclusions

To our best knowledge, this is the first study exploring the specific clinical impact of PA therapy in ADL and the self-awareness of USN for subacute stroke patients during rehabilitation. The results suggested that the improvement of oculomotor exploration by PA may generalize the behavioral level in a daily living environment. In addition, the study suggested that PA could accelerate the self-awareness of neglect and could facilitate motor learning to obtain a higher goal of ADL during subacute rehabilitation.

## Figures and Tables

**Figure 1 brainsci-11-00347-f001:**
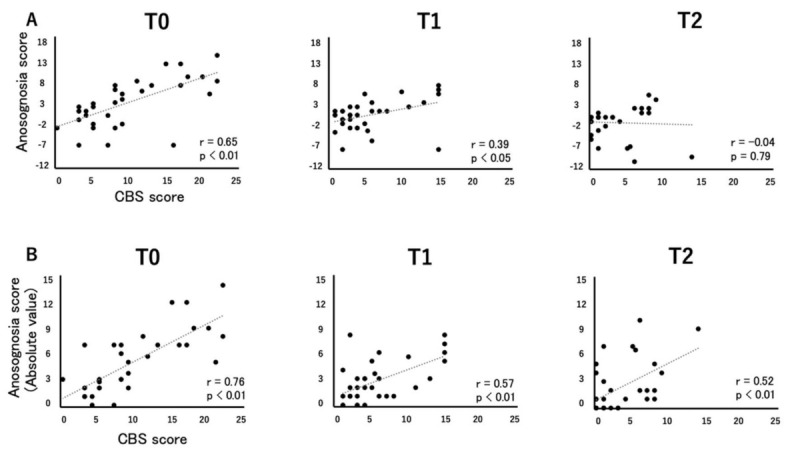
Correlation between Catherine Bergego Scale score and anosognosia score. T0 = Baseline; T1 = After treatment; T2 = Follow-up. (**A**) Correlation between CBS score and Anosognosia score of the control and prism groups. (**B**) Correlation between CBS score and Anosognosia score (absolute value) of the control and prism groups.

**Figure 2 brainsci-11-00347-f002:**
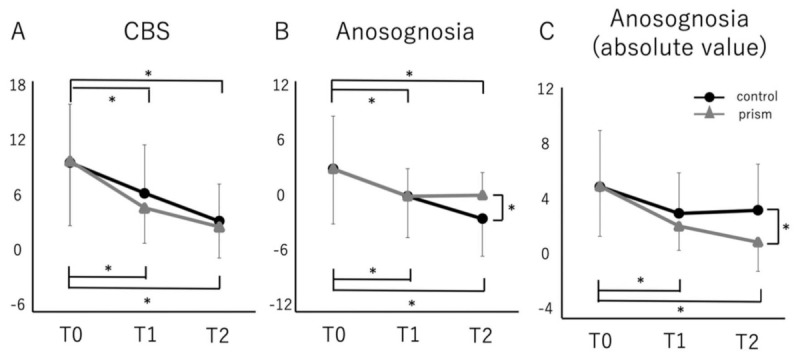
Comparison of Catherine Bergego Scale score and anosognosia score (and absolute value) between control and prism groups. Comparison of the Catherine Bergego Scale score (**A**), Anosognosia score (**B**), and absolute value of Anosognosia score (**C**) between control (black line) and prism groups (gray line): an error bar indicates standard deviation. The CBS, anosognosia and absolute values were rated at baseline (T0) and just after treatment (T1) and follow-up (T2). Error bars indicate standard deviation. * *p* < 0.05.

**Table 1 brainsci-11-00347-t001:** Baseline Characteristics of the two groups.

Characteristic	Control	Prism
	(*n* = 19)	(*n* = 15)
Male/female	14/5	11/4
Age, year	66.5 ± 7.7	64 ± 11.5
Time post stroke onset in days	27.1 ± 14.2	19.6 ± 5.78
Days between onset and intervention	64.4 ± 20.9	67.1 ± 18.4
Length of stay	138.3 ± 43.0	127 ± 42.2
Discharge destination (Home/hospital or nursing home)	12/6	10/5
CBS		
CBS; maximum, 30	9.6 ± 6.1	9.7 ± 6.8
Self-evaluation;	7.1 ± 6.1	6.8 ± 4.0
maximum, 30		
Anosognosia score;	2.9 ± 5.7	2.9 ± 5.4
maximum, 30		
Anosognosia score (absolute value);	5.0 ± 3.9	5.0 ± 3.4
maximum, 30		

**Table 2 brainsci-11-00347-t002:** Comparison of each item of CBS between prism group and control group. Baseline = T0, After treatment = T1, Follow-up = T2. Values are the means ± SD. * *p* < 0.05 (Mann-Whitney test).

	Prism			Control			*p* Value		
	T0	T1	T2	T0	T1	T2	T0	T1	T2
Grooming	0.74 ± 0.80	0.30 ± 0.59	0.24 ± 0.42	0.42 ± 0.76	0.36 ± 0.68	0.36 ± 0.68	0.17	0.92	0.83
Dressing	1.61 ± 0.90	0.97 ± 0.66	0.64 ± 0.81	1.51 ± 0.96	1.09 ± 0.72	0.99 ± 0.73	0.63	0.58	0.16
Eating	0.34 ± 0.73	0.17 ± 0.52	0.11 ± 0.29	0.31 ± 0.67	0.26 ± 0.56	0.21 ± 0.53	1	0.56	0.77
Mouth cleaning	0.46 ± 0.63	0.24 ± 0.42	0.24 ± 0.42	0.42 ± 0.69	0.31 ± 0.58	0.21 ± 0.41	0.72	0.89	0.79
Gaze orientation	0.8 ± 1.14	0.33 ± 0.61	0.2 ± 0.41	1.05 ± 0.84	0.78 ± 0.78	0.63 ± 0.68	0.3	0.07	0.04 *
Knowledge of left limbs	1.73 ± 0.96	1 ± 0.53	0.8 ± 0.67	1.63 ± 0.89	1.21 ± 0.85	0.78 ± 1.03	0.69	0.44	0.66
Auditory attention	0.73 ± 0.79	0.4 ± 0.50	0.33 ± 0.61	0.94 ± 0.97	0.63 ± 0.68	0.52 ± 0.69	0.57	0.35	0.37
Moving (collisions)	1.6 ± 0.91	0.66 ± 0.48	0.73 ± 0.70	1.51 ± 0.95	1.04 ± 0.76	0.67 ± 0.73	0.81	0.13	0.77
Spatial orientation	0.67 ± 0.98	0.33 ± 0.48	0.13 ± 0.35	1.15 ± 0.95	0.62 ± 0.81	0.46 ± 0.67	0.14	0.37	0.11
Finding personal belongings	1.06 ± 1.03	0.4 ± 0.50	0.13 ± 0.35	1.11 ± 0.99	0.68 ± 0.82	0.63 ± 0.68	0.88	0.39	0.02 *

**Table 3 brainsci-11-00347-t003:** Catherine Bergego Scale score and Anosognosia Score (and absolute value).

	Control			Prism		
	T0	T1	T2	T0	T1	T2
CBS	9.61 ± 6.1	6.38 ± 5.1	3.44 ± 3.9	9.78 ± 6.8	4.82 ± 3.8	2.83 ± 3.3
Anosognosia score	2.91 ± 5.7	0.05 ± 4.3	−2.28 ± 3.9	2.91 ± 5.4	0.06 ± 2.9	0.16 ± 2.4
Anosognosia score (absolute value)	5.02 ± 3.9	3.17 ± 2.8	3.39 ± 3.2	5.04 ± 3.5	2.27 ± 1.7	1.17 ± 2.0

## Data Availability

Data available on request due to restrictions e.g., privacy or ethical. The data presented in this study are available on request from the corresponding author. The data are not publicly available due to data privacy regulations.

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
