# Peer review of "Effect of Prism Adaptation Therapy on the Activities of Daily Living and Awareness for Spatial Neglect: A Secondary Analysis of the Randomized, Controlled Trial"

_brainsci, 2021, doi:10.3390/brainsci11030347_

Round 1
Reviewer 1 Report
This paper adds a novel aspect to the already published evaluation paper on the effects of PA. It is interesting and presents some interesting clinical conclusions.
The editing can be largely improved. As most blatant typo I can indicate line 117 on page 3 in which "right hemispace" is confounded with "right hemisphere", but the whole text can be improved.
On the side of content, I would like the authors to define "the subacute stage" of stroke (in which the patients were tested) more sharply.
I would also indicate significant results in Table 2 with an asterisc, to improve clarity.
Last but not least, I would define re-exploration of right hemispace and re-marking of targets as a perseveration error, that might be explained by a working memory deficit.
Author Response
Response to reviewers’ comments
Reviewer 1
Comments and Suggestions for Authors
This paper adds a novel aspect to the already published evaluation paper on the effects of PA. It is interesting and presents some interesting clinical conclusions.
A: We thank you for carefully reading our manuscript and pointing out some of its problems. We have tried to improve our manuscript according to the comments addressed below.
The editing can be largely improved. As most blatant typo I can indicate line 117 on page 3 in which "right hemispace" is confounded with "right hemisphere", but the whole text can be improved.
A: I apologize for many typos and grammatical errors though the original manuscript was proofed by a professional English proof reader before submission. I checked our manuscript carefully and correct these errors. In addition, the final version of revised manuscript was edited by a professional English proof reader again.
On the side of content, I would like the authors to define "the subacute stage" of stroke (in which the patients were tested) more sharply.
A: We described the definition of “the subacute stage” in this study. (line 88-90 on Page 2)
I would also indicate significant results in Table 2 with an asterisc, to improve clarity.
A: We added asterisk to significant result in Table 2.
Last but not least, I would define re-exploration of right hemispace and re-marking of targets as a perseveration error, that might be explained by a working memory deficit.
A: We agree your opinion. We described about that in discussion. (line210-211 on Page 7)

Reviewer 2 Report
I read with interest the paper by Mizuno et al, in which they re-analyze in detail previous RCT data. The manuscript is well written, well-structured and it reports the correct number of references. Overall the study makes some clarification on aspects ameliorated by USN and other aspects that did not take advantage of PA therapy. There are only some little fixes and one general question that need to be answered.
General
In the discussion, the results were explained by the impaired working memory. I suggest reporting the premotor theory that could explain some effect of PA. Prism adaptation has been shown to have a strong effect on premotor neglect. See Saevarsson, Kristjnsson, 2013; Striemer, Danckert, 2010; Saevarsson et al., 2014; Facchin et al., 2019.
Minor
In the 2.2 paragraph, please specify who performed the evaluation and clarify the observation made for the scoring of CBS by different figures during care (occupational therapist, neuropsychologist, physician, etc.).
Line 154: “score” instead of “sore”.
Line 180-183 The text reported seems to be the figure caption and not the text of the manuscript, please check or fix it.
Line 237, an alternative view could be that prism adaptation could boost the rehabilitation process.
The author contribution paragraph is not separated from the conclusion paragraph
References suggested for introduction
Di Monaco, M., Schintu, S., Dotta, M., Barba, S., Tappero, R., & Gindri, P. (2011). Severity of unilateral spatial neglect is an independent predictor of functional outcome after acute inpatient rehabilitation in individuals with right hemispheric stroke. Archives of physical medicine and rehabilitation, 92(8), 1250-1256.
Ronchi, R., Bolognini, N., Gallucci, M., Chiapella, L., Algeri, L., Spada, M. S., & Vallar, G. (2014). (Un) awareness of unilateral spatial neglect: A quantitative evaluation of performance in visuo-spatial tasks. cortex, 61, 167-182.
References suggested for premotor explanation
Saevarsson, S., & Kristjánsson, Á. (2013). A note on Striemer and Danckert's theory of prism adaptation in unilateral neglect. Frontiers in human neuroscience, 7, 44.
Striemer, C. L., & Danckert, J. (2010). Dissociating perceptual and motor effects of prism adaptation in neglect. Neuroreport, 21(6), 436-441.
Saevarsson, S., Eger, S., & Gutierrez-Herrera, M. (2014). Neglected premotor neglect. Frontiers in human neuroscience, 8, 778.
Facchin, A., Sartori, E., Luisetti, C., De Galeazzi, A., & Beschin, N. (2019). Effect of prism adaptation on neglect hemianesthesia. Cortex, 113, 298-311.
Author Response
Response to reviewers’ comments
Reviewer 2
Comments and Suggestions for Authors
I read with interest the paper by Mizuno et al, in which they re-analyze in detail previous RCT data. The manuscript is well written, well-structured and it reports the correct number of references. Overall the study makes some clarification on aspects ameliorated by USN and other aspects that did not take advantage of PA therapy. There are only some little fixes and one general question that need to be answered.
General
In the discussion, the results were explained by the impaired working memory. I suggest reporting the premotor theory that could explain some effect of PA. Prism adaptation has been shown to have a strong effect on premotor neglect. See Saevarsson, Kristjnsson, 2013; Striemer, Danckert, 2010; Saevarsson et al., 2014; Facchin et al., 2019.
- Thank you for your important suggestion. We described the opinion in discussion (line 203-207 on Page 7).
Minor
In the 2.2 paragraph, please specify who performed the evaluation and clarify the observation made for the scoring of CBS by different figures during care (occupational therapist, neuropsychologist, physician, etc.).
A: In this study, The CBS scored by an occupational therapist. (line 124-125 on Page 3)
Line 154: “score” instead of “sore”.
A: We corrected the typo.
Line 180-183 The text reported seems to be the figure caption and not the text of the manuscript, please check or fix it.
A: We moved it to figure caption.
Line 237, an alternative view could be that prism adaptation could boost the rehabilitation process.
A: We agree your opinion. We described it in discussion (243-245).
The author contribution paragraph is not separated from the conclusion paragraph
A: We separated them.
References suggested for introduction
Di Monaco, M., Schintu, S., Dotta, M., Barba, S., Tappero, R., & Gindri, P. (2011). Severity of unilateral spatial neglect is an independent predictor of functional outcome after acute inpatient rehabilitation in individuals with right hemispheric stroke. Archives of physical medicine and rehabilitation, 92(8), 1250-1256.
Ronchi, R., Bolognini, N., Gallucci, M., Chiapella, L., Algeri, L., Spada, M. S., & Vallar, G. (2014). (Un) awareness of unilateral spatial neglect: A quantitative evaluation of performance in visuo-spatial tasks. cortex, 61, 167-182.
A: We added these references in introduction,
References suggested for premotor explanation
Saevarsson, S., & Kristjánsson, Á. (2013). A note on Striemer and Danckert's theory of prism adaptation in unilateral neglect. Frontiers in human neuroscience, 7, 44.
Striemer, C. L., & Danckert, J. (2010). Dissociating perceptual and motor effects of prism adaptation in neglect. Neuroreport, 21(6), 436-441.
Saevarsson, S., Eger, S., & Gutierrez-Herrera, M. (2014). Neglected premotor neglect. Frontiers in human neuroscience, 8, 778.
Facchin, A., Sartori, E., Luisetti, C., De Galeazzi, A., & Beschin, N. (2019). Effect of prism adaptation on neglect hemianesthesia. Cortex, 113, 298-311.
A: We added these references in discussion.
